

# Geopolitical species revisited: genomic and morphological data indicate that the roundtail chub *Gila robusta* species complex (Teleostei, Cyprinidae) is a single species

Joshua M. Copus[1], W. L. Montgomery[2], Zac H. Forsman[1], Brian W. Bowen[1] and Robert J. Toonen[1]

[1] Hawaii Institute of Marine Biology, University of Hawaii at Manoa, Kaneohe, HI, USA
[2] Department of Biology, Northern Arizona University, Flagstaff, AZ, USA

## ABSTRACT

The *Gila robusta* species complex in the Lower Colorado River Basin has a complicated taxonomic history. Recent authors have separated this group into three nominal taxa, *G. robusta*, *G. intermedia*, and *G. nigra*, however aside from location, no reliable method of distinguishing individuals of these species currently exists. To assess relationships within this group, we examined morphology of type specimens and fresh material, and used RADseq methods to assess phylogenetic relationship among these nominal species. Maximum likelihood and Bayesian inference tree building methods reveal high concordance between tree topologies based on the mitochondrial and nuclear datasets. Coalescent SNAPP analysis resolved a similar tree topology. Neither morphological nor molecular data reveal diagnostic differences between these species as currently defined. As such, *G. intermedia* and *G. nigra* should be considered synonyms of the senior *G. robusta*. We hypothesize that climate driven wet and dry cycles have led to periodic isolation of population subunits and subsequent local divergence followed by reestablished connectivity and mixing. Management plans should therefore focus on retaining genetic variability and viability of geographic populations to preserve adaptability to changing climate conditions.

## INTRODUCTION

The fish genus *Gila* Baird & Girard 1853a contains 20 currently recognized species in the Western United States and Mexico. Of these, *G. cypha* Miller 1946, *G. elegans* Baird & Girard 1853a, *G. intermedia* (Girard 1856), *G. jordani* Tanner 1950, *G. nigra* Cope & Yarrow 1875, *G. robusta* Baird & Girard 1853a, and *G. seminuda* Cope in Cope & Yarrow 1875 inhabit the Colorado River Basin and make up the *Gila robusta* species complex (Gerber, Tibbets & Dowling, 2001). The Lower Colorado River Basin, separated from the

Corresponding author
Joshua M. Copus, jcopus@hawaii.edu

Upper Colorado River Basin by Glen Canyon Dam, is occupied by *G. robusta*, *G. intermedia*, and *G. nigra*. These *Gila* populations, as with many freshwater fishes within the Lower Colorado River Drainage are in decline from anthropogenic threats such as habitat destruction and modification accompanying human population growth and interactions with non-native fishes (*Minckley & Marsh, 2009*). These declines led to the listing of *G. intermedia* as endangered under the US Endangered Species Act (*US Fish and Wildlife Service (USFWS), 2005*) and a proposal for *G. robusta* and *G. nigra* to be listed as threatened (*US Fish and Wildlife Service (USFWS), 2015*).

Over the past 150 years, the *G. robusta* complex within the Lower Colorado River Basin has received considerable attention in an attempt to resolve relationships among the populations that inhabit these drainages. Hypotheses such as ecophenotypic plasticity, introgression, and cryptic speciation have all been invoked to account for geographic variation in genetic structure, as well as morphological and ecological traits within and among species of *Gila* (*Miller, 1946*; *Dowling & DeMarais, 1993*; *Gerber, Tibbets & Dowling, 2001*; *Marsh, Clarkson & Dowling, 2017*). Numerous molecular studies have attempted to resolve the relationships of the *Gila robusta* complex. Although interpretations of the data vary, there is no clear evidence to date that the three nominal species of *Gila* in the drainages of the Lower Colorado River basin represent reproductively isolated and distinct evolutionary units (*DeMarais, 1992*; *DeMarais et al., 1992*; *Dowling & DeMarais, 1993*; *Schönhuth et al., 2012*, *2014*; *Dowling et al., 2015*; *Marsh, Clarkson & Dowling, 2017*), nor has a reliable method (morphological or molecular) of assigning individual fish to species been identified (*Moran et al., 2017*; *Carter et al., 2018*). The current practice of species identification for managers and researchers working with the *G. robusta* complex requires identifications based on collection locality as determined by *Rinne (1969)* and later revised by *Minckley & DeMarais (2000)* based on mean morphological differences of populations rather than diagnosable morphological or molecular characters of individuals, because no such characters have been identified. The *Gila* populations within these localities were assigned to distinct species (*G. robusta* and *G. intermedia*) and subspecies (*G. robusta grahami* (=*nigra*); *Rinne, 1969*, *1976*) based on mean morphological differences among populations. Rinne interpreted these mean differences to represent distinct taxonomic units (species and subspecies) but dismissed the variance in morphology that exists within each population. *Minckley & DeMarais (2000)* revised the geographic ranges and taxonomic status of this group and developed a taxonomic key for the identification of what they recognized as three distinct species. Unfortunately, the implementation of this key has been problematic for managers and researchers attempting to assign species names to individual fish, in part due to the reliance on population means as diagnostic characters, and in part due to confusion arising from variation among individuals within and among locations.

A recent review of the specific status of *Gila intermedia* and *Gila nigra* as distinct from *Gila robusta* was undertaken by the American Society of Ichthyologists and Herptologists— American Fisheries Society (ASIH-AFS) Committee on the names of fishes. Upon reviewing all available literature and data, including much of the data presented here, and hosting a symposium on the topic, the committee concluded that there was no evidence
that *G. intermedia* and *G. nigra* were distinct from *G. robusta* (*Page et al., 2017*). Since this decision from the ASIH-AFS, the USFWS has withdrawn their proposal to list *G. robusta* and *G. nigra* as threatened under the Endangered Species Act until a species status assessment of the newly defined *G. robusta* taxon can be undertaken.

Here, we review the systematic and taxonomic history of this group to provide a foundation for evaluating the nomenclature within the *G. robusta* complex. We provide a comprehensive morphological comparison of the type series of each species, as well as fresh material from streams throughout the Lower Colorado River Basin, to test for diagnostic characters. Finally, we use molecular phylogenomics to ascertain the evolutionary relationships among populations and putative species.

## Systematic and taxonomic review

There has been considerable confusion regarding the systematics and taxonomy of the *Gila robusta* complex within the Lower Colorado River Basin, largely due to a complex array of phenotypes (*DeMarais et al., 1992*). Many species within the genus *Gila Baird & Girard 1853a* have been described multiple times and the *G. robusta* complex of the Lower Colorado River Basin is no exception. We identified fifteen specific names and seven generic names applied to these fishes.

### G. robusta Baird & Girard 1853a

*Gila robusta Baird & Girard 1853a* is the type species for the genus. The type series of the species is cataloged as USNM 246, but a note included with the type specimens states, "These, the types of *Gila robusta* B.+G., are cat. No. 246. They were reentered by error as 47,983 and attributed to nos. 276 + 273, which are cods! Nos. 276 + 277 were attributed as type nos. of this species, by error, by *Jordan & Evermann (1896*: 227*)*. R.R. Miller III: 1945" (Fig. S1). This is one of a number of clerical errors uncovered in the taxonomic history of this fish and is indicative of the historic pattern of confusion surrounding the systematics of this complex.

The original description reported that the collection locality for the syntypes was the Zuni River, New Mexico, but *Smith, Miller & Sable (1979)* suggest that this locality is a clerical error, based on the argument that the Zuni River was unsuitable habitat for *G. robusta* during the time at which the type specimens were collected (1851). *Smith, Miller & Sable (1979)* suggested that the specimens were actually from another collection site of the Sitgreaves expedition, the Little Colorado River, below Grand Falls, Coconino County, Arizona. *Sublette, Hatch & Sublette (1990)* dispute the assertions of *Smith, Miller & Sable (1979)*, and contend that the syntypes were collected from the Zuni River in 1851, and note that additional specimens were subsequently collected twice in 1873 and once in 1879 on the Zuni River by different collectors. It is highly unlikely that multiple clerical errors on different expeditions would have occurred at this locality with this species. *Sublette, Hatch & Sublette (1990)* suggest that the Zuni River represented marginal habitat that may have received recruits from the Rio Pescado and that they have since been extirpated from the Zuni River. The journal of SW Woodhouse, the naturalist who originally collected these fishes, clearly states that on

Saturday the 6th of September 1851, he received these fishes while camped at the Zuni Pueblo, which were "collected from the creek." The argument that the syntypes were collected on the Little Colorado River rather than the Zuni is therefore dubious and we maintain the type locality as the Zuni River for *G. robusta*.

### G. intermedia (Girard 1856)

*Girard (1856)* described *Tigoma intermedia* from specimens collected on the Rio San Pedro in Arizona and noted that it was morphologically intermediate between *T. pulchella* (*Baird & Girard 1854*) and *T. purpurea Girard 1856* (both of which are now regarded as valid species within the genus *Gila*). *Evermann & Rutter (1895)* placed *T. intermedia* within the genus *Leuciscus Cuvier 1816*. *Jordan & Evermann (1896)* synonymized *Tigoma* and *Richardsonius Girard 1856* with *Leuciscus*, and suggested that *L. intermedius* and *L. niger* (Cope in *Cope & Yarrow 1875*) (now *Gila nigra* Cope in *Cope & Yarrow 1875*) may be conspecific. *Fowler (1924)* retained this genus placement for *L. intermedius*. *Snyder (1915)* regarded *T. intermedia* and *G. nigra* as synonyms of *Gila gibbosa Baird & Girard 1854*, which he placed within the genus *Richardsonius Girard 1856*. *Jordan & Gilbert (1883)* asserted that *gibbosus Baird & Girard 1854* was unavailable due to homonymy resulting from placement of *Leuciscus gibbosus Ayres 1854* and *Gila gibbosa Baird & Girard 1854* within the genus *Squalius Bonaparte 1837* (*Gila gibbosa Baird & Girard 1854* being the junior homonym). *Jordan & Evermann (1896)* placed both of these species within the genus *Leuciscus Cuvier 1816*, creating another case of homonymy, this time within *Leuciscus*. In both cases, *gibbosus Ayres 1854* was identified as the senior homonym; however, that name was itself preoccupied by *Leuciscus* between *gibbosus Storer 1845* and *gibbosus Ayres 1854*. In any case, *G. gibbosa Baird & Girard 1854* is not an available name. *Miller (1945)* treated *intermedia* as a subspecies of *G. robusta Baird & Girard 1853a*, which was followed by subsequent authors (*Miller, 1946*, *1961*; *Uyeno, 1961*; *La Rivers, 1962*; *Sigler & Miller, 1963*; *Miller & Lowe, 1964*, *1967*; *Uyeno & Miller, 1965*; *Barber & Minckley, 1966*; *Cole, 1968*; *Minckley & Alger, 1968*; *Minckley, 1969*; *Lee et al., 1980*; *Robins, Bailey & Bond, 1980*).

*Rinne (1969)* recognized two distinct species within the Lower Colorado River Basin: *G. robusta* for the more broadly distributed form, and *G. intermedia* for populations principally distributed in central and southern Arizona. Within the synonymy of *G. intermedia*, he also included *G. gibbosus*, *G. nigra*, and *G. lemmoni*, noting that the former was unavailable due to homonymy, thereby asserting *G. intermedia* as the next available name. *Rinne (1969)* does not appear to have examined any of the type series of *G. intermedia*, but states "in all respects, they (populations he labels as *G. intermedia*) correspond to the original description of *Tigoma* (=*Gila*) *intermedia Girard 1856*." The original description of *G. intermedia* (*Girard 1856*) consisted of the following text: "Intermediate between *T. pulchella* and *T. purpurea*, more closely related however to the former than to the latter. The fins are much less developed, the inferior fins especially are quite small." The original description did not include enough diagnostic characters to confidently conclude that the populations *Rinne (1969)* defines as *G. intermedia* are conspecific with the type specimens of *G. intermedia*, and there is no evidence that Rinne examined either *pulchella* or *purpurea* before asserting that *G. intermedia* was in fact,

the correct name for these populations. Nevertheless, most subsequent authors followed *Rinne (1969)* in treating *G. intermedia* as a valid species (*Stout, Bloom & Glass, 1970*; *Minckley, 1971*, *1973*; *Rinne, 1976*; *Hocutt & Wiley, 1986*; *Minckley, Hendrickson & Bond, 1986*; *Sublette, Hatch & Sublette, 1990*; *Robins, 1991*; *Winfield & Nelson, 1991*; *Espinosa-Pérez, Gaspar-Dillanes & Fuentes-Mata, 1993*; *Gilbert, 1998*; *Minckley & DeMarais, 2000*; *Norris, Fischer & Minckley, 2003*; *Nelson et al., 2004*; *Scharpf, 2005*; *Miller, 2005*; *Minckley & Marsh, 2009*; *Page & Burr, 2011*; *Page et al., 2013*), despite the known complex taxonomic history and a lack of any discrete identifying characteristics.

### G. grahamii Baird & Girard 1853b

Although *Gila grahamii* is currently recognized as a synonym of *G. robusta*, due to the complicated systematic history of this group, particularly with respect to *G. nigra*, we include this nomenclatural account to promote clarity. *Baird & Girard (1853b)* described *G. grahamii* (often misspelled in the literature as *grahami*) from specimens collected in the Rio San Pedro, Gila River basin. *Günther (1868)* placed it within the genus *Leuciscus Cuvier 1816*. *Cope & Yarrow (1875)* placed the species back in the genus *Gila* and recognized it as distinct from both *G. robusta Baird & Girard 1853a* and *G. nigra* Cope in *Cope & Yarrow 1875*. *Evermann & Rutter (1895)* treated *G. grahamii* as a synonym of *G. robusta*, and subsequent early authors followed this assignment. This synonymy remained stable until *Rinne (1969)*, who regarded *grahamii* as a subspecies of *G. robusta*, recognized populations collected from the tributaries of the Verde River and the upper Gila River system as distinct from the subspecies *G. r. robusta* in the main stem Verde and Gila Rivers. *Rinne (1969)* recognized current San Pedro populations (type locality of *G. grahamii*) as belonging to the species he referred to as *G. intermedia*, even though he accepted *G. r. grahamii* for his "headwater" form (again, apparently without examining type specimens). In the years between 1969 and 2000, there was not consistent recognition of *G. r. grahamii* as a valid subspecies, but no authors treated it as a valid species, or as a synonym or subspecies of any species other than *G. robusta* (*Rinne, 1976*; *Lee et al., 1980*; *Robins, Bailey & Bond, 1980*; *Holden & Minckley, 1980*; *Mayden, 1992*; *La Rivers, 1994*; *Gilbert, 1998*). *Minckley & DeMarais (2000)* regarded the populations referred to by *Rinne (1969)* as *G. r. grahamii* as representing a distinct species, even though they noted that it is likely of hybrid origin and paraphyletic. However, they also noted that the syntypes of *G. grahamii* belong to what *Rinne (1969)* regarded as the subspecies *G. r. robusta* (citing personal communication between RR Miller & WL Minckley). Therefore, they recognized *G. nigra* Cope in *Cope & Yarrow 1875*, as the earliest available name for the species previously referred to by *Rinne (1969)* as *G. r. grahamii*.

### G. nigra Cope 1875

*Gila nigra* Cope in *Cope & Yarrow (1875)* was described from specimens collected in Ash Creek and at San Carlos, Arizona. *Jordan & Gilbert (1883)* placed it in the genus *Squalius Bonaparte 1837*, and later *Jordan & Evermann (1896)* placed it in the genus *Leuciscus*. They also regarded *Gila gibbosa Baird & Girard 1854* as conspecific, but unavailable due to homonymy (see above). *Gilbert & Scofield (1898)* synonymized *G. nigra* with *T. intermedia*, which they placed in the genus *Leuciscus*. *Snyder (1915)* regarded *G. niger* as

a synonym of *G. gibbosa* (within the genus *Richardsonius*), failing to recognize that the latter species name was not available due to homonymy. *Fowler (1924)* followed *Gilbert & Scofield (1898)* in treating *G. nigra* as a synonym of *intermedius*, within the genus *Leuciscus*. Subsequent treatments placed *nigra* in synonymy with *intermedia* (see above). This synonymy was broadly followed until *Minckley & DeMarais (2000)* recognized *G. nigra* as the earliest available name to refer to the species treated by *Rinne (1969)* as *G. r. grahamii*.

***Synonymies now considered valid***

Along with the considerable synonymy of *Gila robusta* are species that were at one time considered synonyms of *G. robusta* and are now considered valid. *Gila elegans Baird & Girard 1853a* was treated as a synonym of *G. robusta* by *Ellis (1914)* and as a subspecies by *Miller (1945)* and *La Rivers (1994)*. However, *G. elegans* was treated as valid by *Vanicek (1967)* and subsequent authors. *Gila jordani Tanner 1950* was treated as a subspecies of *G. robusta* by *Rinne (1976)*, *Lee et al. (1980)*, *La Rivers (1994)*, and *Gilbert (1998)* but was recognized as valid by *Minckley & Marsh (2009)* and subsequent authors. *Clinostomus pandora Cope 1872* was treated as a synonym of *G. robusta* by *Ellis (1914)* but valid as *Gila pandora* (*Cope 1872*) by subsequent authors. *Tigoma egregia Girard 1858* was treated as a synonym of *G. robusta* by *Ellis (1914)* but regarded as valid as *Richardsonius Girard 1856 egregius* (*Girard 1858*) by subsequent authors. Finally, *Gila seminuda Cope & Yarrow 1875* was treated as a subspecies of *G. robusta* by *Ellis (1914)*, *Snyder (1915)*, *Rinne (1976)*, and *Lee et al. (1980)* but regarded as valid by *Gilbert (1998)* and subsequent authors.

## MATERIALS AND METHODS

Type material for each of the currently recognized species of the *G. robusta* complex as well as the type of the *G. robusta* synonym *G. grahamii* were obtained from the Smithsonian National Museum of Natural History (*G. robusta* (USNM 246, *N* = 2), *G. nigra* (USNM 16972, *N* = 3; 16987 *N* = 2), *G. intermedia* (USNM 232, *N* = 4), and *G. grahamii* (USNM 253, *N* = 1)).

Due to the problems associated with the taxonomic key (*Moran et al., 2017*; *Carter et al., 2018*) the current practice of species identification for managers and researchers working with the *G. robusta* complex within the Lower Colorado River Basin is based on drainage location, as assigned by *Rinne (1969)* and later revised by *Minckley & DeMarais (2000)*. We follow this convention because no alternative method of assignment is currently available. Morphological and molecular analysis of fresh specimens of each nominal species (*G. robusta N* = 6, *G. intermedia N* = 6, and *G. nigra N* = 5) as well as *G. elegans* (*N* = 1) and *G. cypha* (*N* = 1) as out-groups were analyzed from streams throughout their range, with the exception of O'Donnell Canyon material collected from the captive population held at the International Wildlife Museum, and Eagle Creek and Verde River samples collected from the Bubbling Ponds Fish Hatchery. One individual per location across the range of each species was analyzed to capture as much within species variation as possible (Fig. 1) Tissue of *G. elegans* and *G. cypha* were obtained from

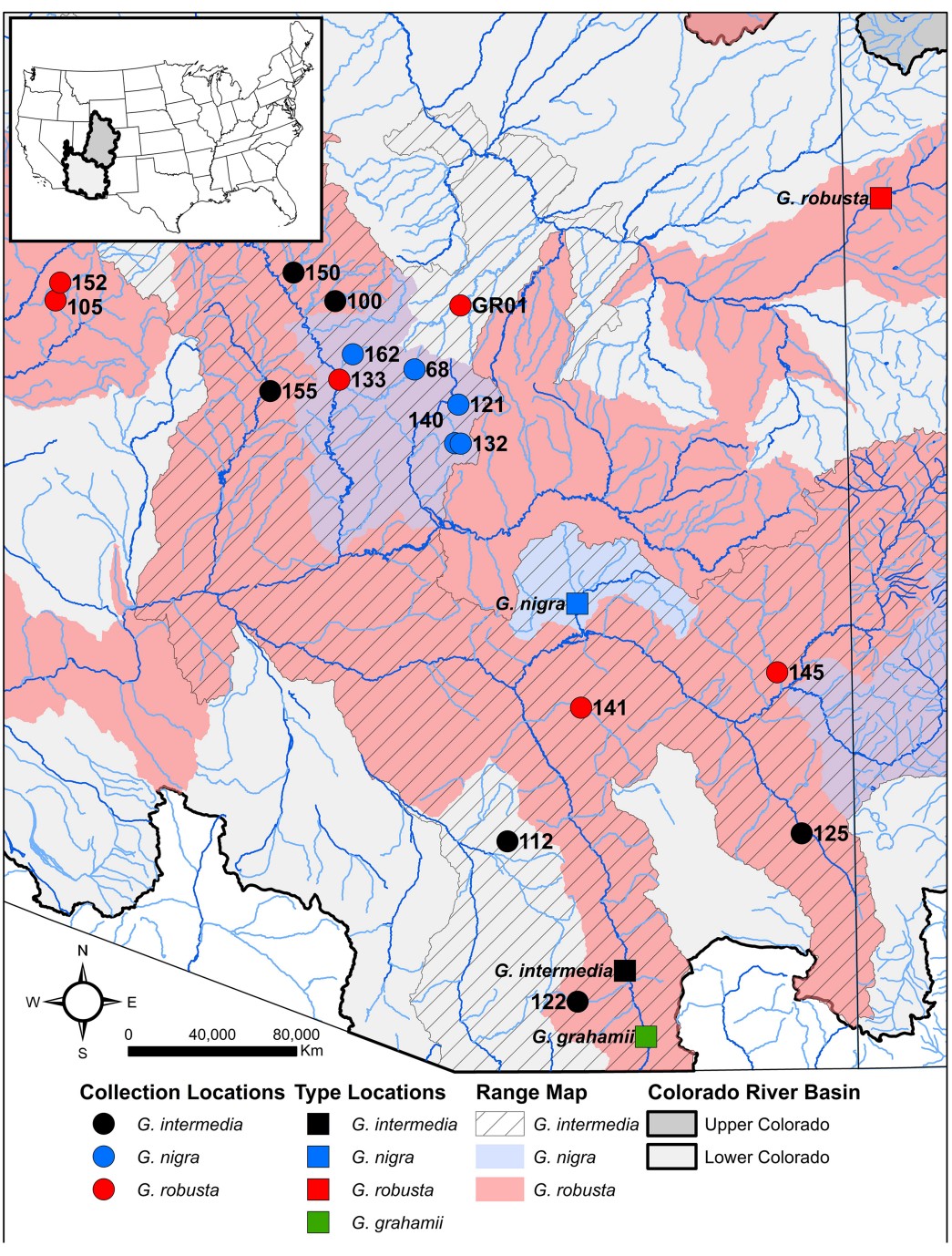

**Figure 1 Map of collecting locations.** Circles indicate collecting location of fresh material; squares represent locality of type series for *G. robusta* (red), *G. intermedia* (black), *G. nigra* (blue), and *G. grahamii* (green). Collecting locations of type material are approximations based on original reports. Gray shaded areas with black outline indicate the Upper Colorado River basin (dark gray) and Lower Colorado River Basin (light gray) and currently recognized ranges based on IUCN Red List maps are for *G. robusta* (red), *G. nigra* (blue), and *G. intermedia* (black diagonal stripes). 🖼 DOI: 10.7717/peerj.5605/fig-1

the Arizona Game and Fish Department Bubbling Ponds Fish Hatchery. Tissue samples were stored in both salt-saturated DMSO solution (20% dimethyl sulfoxide, 0.25 M EDTA, pH 8.0, saturated with NaCl; *Seutin, White & Boag, 1991*; *Gaither et al., 2011*) and RNA Later (Thermo Fisher Scientific, Waltham, MA, USA) for reduced representation genomic sequencing using the ezRAD protocol (*Toonen et al., 2013*; *Knapp et al., 2016*). Specimens were frozen prior to morphological analyses.

## Morphological analysis

Fresh specimens were thawed and radiographed to aid in morphological analysis. Species names were assigned to fresh specimens based on sampling location as designated by *Rinne (1969)* and *Minckley & DeMarais (2000)* (see above). Meristic and morphometric analysis follow methods outlined in *Hubbs & Lagler (1958)*. All statistical comparisons of morphology were implemented in R (*R Core Team, 2016*). Linear regression was performed on each of the morphometric characters to test for allometric growth and confirm that each of these characters scale linearly with size. Variables were then standardized by length for comparison. *F*-tests (to determine equal or unequal variance) and *t*-tests were performed on all standardized variables to test whether shrinkage due to preservation causes significant differences between type material and fresh material for any variable. We find that allometric growth and shrinkage from preservation are not significant between fresh and preserved specimens and consequently comparisons between size and preservation method were possible (Table S1). Morphometric measurements are presented as a percentage of standard length, head length, or body depth.

## Reduced representation genomic data production

Genomic DNA was extracted from tissue using the Omega E.Z.N.A Tissue DNA Kit (Omega Biotek, Norcross, GA, USA) following the manufacturer's protocol except that HPLC grade $H_2O$ was substituted for the elution buffer. DNA aliquots were visualized following electrophoresis on a 1% agarose gel to assess quantity and quality. For extractions that did not yield >1 µg high molecular weight DNA (>10 kb), multiple extractions from the same individual were completed and extractions were pooled and concentrated using an Eppendorf Vacufuge plus (Eppendorf, Hauppauge, NY, USA). Extractions were quantified with AccuBlue (Biotium, Inc., Hayward, CA, USA) high sensitivity dsDNA quantification kit and measured on a SpectraMax M2 microplate reader (Molecular Devices, Sunnyvale, CA, USA). All extractions were subsequently stored at −20 °C until used for library preparation as outlined below.

Size-selected reduced representation genomic libraries were generated following the ezRAD protocol (*Toonen et al., 2013*; *Knapp et al., 2016*). In brief, tissue samples were cleaned with Agencourt AMPure XP beads (Beckman Coulter, Indianapolis, IN, USA) following manufacturers' protocols. High molecular weight DNA was digested using the restriction enzyme *Dpn*II to cleave at all GATC cut sites using a final master mix volume of 25 µl (five µl Buffer, 19 µl HPLC grade $H_2O$, one µl one unit *Dpn*II) to 25 µl dsDNA (one µg). Samples were incubated for 3 h at 37 °C followed by 20 min at 65 °C to denature the enzyme. Following digestion, samples underwent a second bead cleaning

with AMPure XP beads. Library preparation for Illumina sequencing was completed with the KAPA HyperPrep kit (Kapa Biosystems, Wilmington, MA, USA) following manufacturers' protocols. All libraries were size selected to retain 300–500 base pair (bp) fragments and passed through quality control steps (bioanalyzer and qPCR) prior to high-throughput sequencing. Illumina paired-end fragments were sequenced at the Hawaii Institute of Marine Biology Genetics Core Facility using Illumina v3 $2 \times 300$ reads on the MiSeq genomic analyzer (Illumina, San Diego, CA, USA).

## Genomic analysis

### Mitochondrial genome

Raw Illumina reads were paired, trimmed and mapped to the mitochondrial genome of *G. robusta* (GenBank DQ536424.1) using Geneious v.6.1.8 (*Kearse et al., 2012*). Five mapping iterations were completed at high sensitivity. For each sample, consensus sequences of all contigs that successfully mapped to the reference genome were extracted, and all consensus sequences were aligned, manually inspected and low coverage regions removed in Geneious.

### Reduced representation nuclear genome

Reads were trimmed, assembled, and genotyped using the dDocent pipeline (*Puritz, Hollenbeck & Gold, 2014*). Any loci appearing in less than 85% of individuals were excluded from these analyses. The paired-end overlapping read algorithm was used for de novo assembly. Clustering similarity of 0.9 and mapping parameters of A (match score) = 2, B (mismatch score) = 3, and O (gap penalty) = 4 were used. dDocent processing recovered 89,896 loci with an average read depth of 227 in all 19 individuals. Complex variants were decomposed using vcflib (*Garrison & Marth, 2012*) to deconstruct haplotypes and insertions and deletions were removed using VCFtools (*Danecek et al., 2011*). Any contigs that mapped to the mtDNA were removed from the dataset using VCFtools, and the remaining contigs were considered to represent only the nuclear genome. The resulting data were then collapsed into haplotypes with the rad_haplotyper pipeline (https://github.com/chollenbeck/rad_haplotyper; *Willis et al., 2017*), which uses read alignments to record combinations of SNPs present across paired-end reads. For each individual, rad_haplotyper removes complex loci, missing data, paralogs, and sequencing errors. Any locus that is not present in at least 14 of 19 individuals with a depth of coverage of at least 20 is not included in the final dataset. The rad_haplotyper method was employed to overcome many of the problems that can arise with SNP data in the absence of a reference genome, such as inflated homozygosity, artifacts, or inflated heterozygosity (*Willis et al., 2017*). Contigs were then collapsed into genotypes for final analyses. PGDspider v.2.1.1.3 (*Lischer & Excoffier, 2011*) was used to convert the dataset to the required file types for further analysis.

### Phylogenetic analysis

The optimal model of sequence evolution was selected using the Akaike Information Criterion in JModelTest v.2.1.4 (*Posada, 2008*). GTR + G was found to be the best-fit substitution model for both the mtDNA and total evidence datasets. The HKY model was found to be the best-fit model for the nuclear dataset. To calculate posterior probabilities of

clades, MrBayes v.3.2.6 (*Huelsenbeck & Ronquist, 2001*; *Ronquist & Huelsenbeck, 2003*) was used to run a 1,000,000-generation Markov chain implementing the best-fit model for the dataset. We used flat Dirichlet prior probability densities with an initial burn in of 250,000 generations. Trees were saved every 500 generations for a total sample size of 1,500 trees. A majority rule consensus tree calculated from the 2,000 sampled trees was used to determine the posterior probabilities of clades. Under these parameters standard deviations between independent runs stabilized and were all less than 0.01. Maximum likelihood (ML) analyses were conducted using the randomized accelerated maximum likelihood (RAxML) software v.8 (*Stamatakis, 2014*). Best-fit models and 30,000 bootstrap replicates were implemented for all datasets. Uncorrected pairwise divergence times for mtDNA were estimated using Mega v.7.0 (*Kumar, Stecher & Tamura, 2016*). Phylogenetic trees were constructed and visualized using FigTree v.1.4.2 (http://tree.bio.ed.ac.uk/software/figtree/). STRUCTURE analysis and Discriminate analysis of principal components (DAPC) were also implemented (see Supplementary Materials).

### SNAPP coalescent analysis

Species trees were estimated from the nuclear dataset with the SNAPP package in BEAST v.2.3.2 (*Bouckaert et al., 2014*). Polymorphic loci were extracted from the rad_haplotyper output and outgroups were removed. BEAUti (*Drummond et al., 2012*) was executed with the following parameters: all taxa were treated as distinct species/populations, the mutation rates u and v were calculated from the data, and default values were used for the exponential priors. The data were then analyzed in BEAST (*Bouckaert et al., 2014*). The MCMC chain was run for 10,000,000 generations, sampling every 1,000 generations with 300,000 preBurnin. The results of this analysis were visualized using DensiTree (*Bouckaert & Heled, 2014*), and presented as a cloudogram.

### Molecular clock estimation

To estimate the time of coalescence, a Bayesian MCMC approach was implemented in BEAST on the mtDNA dataset, under a coalescent constant population model with a strict clock of 2% per million years (*Brown, George & Wilson, 1979*; *Bowen et al., 2001*; *Reece et al., 2010*). Simulations were run with default priors under the GTR + G model of mutation. Simulations were run for 10 million generations, sampling every 1,000 generations following a 30,000 preBurnin. A total of 10 independent runs were computed to ensure convergence and log files were combined using TRACER v.1.7 (*Rambaut et al., 2018*).

### Tests of introgression

Tests of introgression were performed on the total dataset using ABBA–BABA statistics implemented in HybridCheck (*Ward & Van Oosterhout, 2016*) to assess whether introgression with *G. cypha* could be detected within the *G. robusta* clades. To ensure that our data met the assumptions of ancestry ((P1, P2)P3), required for this test, we only compared in-groups (P1, P2) separated by the distinct clades of our phylogenetic trees. Standard ABBA–BABA calculations use Patterson's D to infer introgression, but this statistic may not be sufficient to separate introgression from ancestral population structure. Therefore, we also include an alternative statistic $F_d$ (*Martin, Davey &*

*Jiggins, 2014*), which estimates the fraction of the genome shared through complete introgression between P2 and P3, and P1 and P3 and is not subject to the same biases as the D statistic. We also report the statistical significance ($p$-value) that expressed the deviation from an equal number of ABBA–BABA sites. Significant admixture was determined by Z scores of 3 or higher.

# RESULTS

## Taxonomic treatment

### *Gila robusta* *Baird & Girard 1853a*

*Gila robusta* *Baird & Girard 1853a*. Zuni River, New Mexico. Syntypes: USNM 246 (USNM 47983; 3, plus 1 pharyngeal arch as #2798)

*Gila gracilis* *Baird & Girard 1853a*; (*Evermann & Rutter, 1895*; *Jordan & Evermann, 1896*; *Gilbert & Scofield, 1898*; *Gilbert, 1998*)

*Gila grahamii* *Baird & Girard 1853b*; (*Cope, 1871*; *Evermann & Rutter, 1895*; *Jordan & Evermann, 1896*; *Fowler, 1924*; *La Rivers, 1994* (as subspecies: *Rinne, 1976*; *Lee et al., 1980*; *Gilbert, 1998*))

*Gila gibbosa* *Baird & Girard 1854*; (*Jordan & Gilbert, 1883*; *Evermann & Rutter, 1895*; *Jordan & Evermann, 1896*; *Gilbert & Scofield, 1898*; *Snyder, 1915*; *Fowler, 1924*; *Gilbert, 1998*)

*Tigoma gibbosa* (*Baird & Girard 1854*); (*Girard, 1856*; *Jordan & Gilbert, 1883*; *Evermann & Rutter, 1895*; *Jordan & Evermann, 1896*; *Gilbert & Scofield, 1898*)

*Richardsonius gibbosus* (*Baird & Girard 1854*); (*Snyder, 1915*)

*Tigoma intermedia* *Girard 1856*; (*Evermann & Rutter, 1895*; *Jordan & Evermann, 1896*; *Gilbert & Scofield, 1898*; *Fowler, 1924*; *Snyder, 1915*; *Gilbert, 1998* (as subspecies: *Miller, 1945, 1946*; *Uyeno & Miller, 1965*; *Barber & Minckley, 1966*; *La Rivers, 1994*)) treated as a full species within *Gila* by *Rinne 1969* and most subsequent authors.

*Squalius intermedius* (*Girard 1856*); (*Jordan & Gilbert, 1883*; *Jordan & Evermann, 1896*; *Gilbert & Scofield, 1898*)

*Leuciscus intermedius* (*Girard 1856*); (*Evermann & Rutter, 1895*; *Gilbert & Scofield, 1898*; *Jordan & Evermann, 1896*; *Fowler, 1924*)

*Ptychocheilus vorax* *Girard 1856*; (*Evermann & Rutter, 1895*; *Jordan & Evermann, 1896*; *La Rivers, 1994*)

*Gila affinis* *Abbott 1860*; (*Jordan & Evermann, 1896*; *Fowler, 1924*)

*Leuciscus zunnensis* *Günther 1868*; (*Jordan & Evermann, 1896*; *Gilbert & Scofield, 1898*; *La Rivers, 1994*; *Gilbert, 1998*)

*Leuciscus robustus* (*Baird & Girard 1853a*); (*Günther, 1868*; *Jordan & Evermann, 1896*; *La Rivers, 1994*)

*Leuciscus grahami* Günther 1868; (*Jordan & Evermann, 1896*)

*Gila nacrea* Cope 1871; (*Evermann & Rutter, 1895*; *Jordan & Evermann, 1896*; *Gilbert, 1998*)

*Gila nigra* Cope in *Cope & Yarrow 1875*; (*Jordan & Gilbert, 1883*; *Gilbert & Scofield, 1898*; *Snyder, 1915*; *Fowler, 1924*; *Gilbert, 1998*)

*Leuciscus niger* (Cope in *Cope & Yarrow 1875*); (*Evermann & Rutter, 1895*; *Jordan & Evermann, 1896*; *Gilbert & Scofield, 1898*)

*Squalius niger* (Cope in *Cope & Yarrow 1875*); (*Jordan & Gilbert, 1883*; *Jordan & Evermann, 1896*)

*Squalius nigra* (Cope in *Cope & Yarrow 1875*); (*Gilbert & Scofield, 1898*, misspelling of *S. niger*)

*Squalius lemmoni* Smith 1884; (*Jordan & Evermann, 1896*; *Gilbert & Scofield, 1898*; *Gilbert, 1998*).

## Morphological analysis

### Type material

Examination of the type series of *G. robusta* ($N = 2$), *G. intermedia* ($N = 4$), *G. nigra* ($N = 5$), and *G. grahamii* ($N = 1$) reveal differences between the types within this complex. However, similar or greater morphological dissimilarity was observed for the *G. grahamii* type (now considered a synonym of *G. robusta*; see above) when compared to each of the other type series (Table 1). Furthermore, these differences apply to only these restricted type series and differences were not supported when the fresh material was added (see below). Unfortunately, the taxonomic key (*Minckley & DeMarais, 2000*) used to assign the names *G. intermedia* and *G. nigra* to the populations that *Rinne (1969*, *1976)* and *Minckley & DeMarais (2000)* recognize as distinct species fails to correctly assign the type material to the correct species.

### Fresh material

Analysis of fresh material of specimens assigned to the three species reveals extensive overlap in characters, prohibiting any definable difference between groups (Table 2). There is no single diagnostic character that can be used for species identification of fresh material, with considerable overlap among species in every morphological character. Likewise no suite of characters can distinguish the fresh material by species unambiguously.

### Comparisons of fresh specimens to type material

Morphological comparisons of type material (Table 1) to fresh specimens (Table 2) also failed to resolve the species as currently recognized (Table 3). Type specimens as well as fresh material exhibit as much or more variation within species as between species. As such, it is impossible to assign any of the fresh specimens back to the type material and thus to species, without location information. No character in putative *G. robusta* specimens could be uniformly assigned back to the type of

**Table 1  Morphometrics and meristics of type material in the *G. robusta* complex.**

| | *G. robusta* USNM 246 | *G. nigra* USNM 16972, 16987 | *G. intermedia* USNM 232 | *G. grahamii* USNM 253 |
|---|---|---|---|---|
| Body depth[1] | 5.2–5.4 | 4.1–4.6 | 3.7–4.0 | 4.5 |
| Head length[1] | 3.5–3.6 | 3.5–3.6 | 3.1–3.4 | 3.6 |
| Head width[3] | 1.3–1.4 | 1.4–1.8 | 1.4–1.6 | 1.7 |
| Head depth[3] | 1.1 | 1.2–1.4 | 1.3–1.4 | 1.5 |
| Snout length[2] | 3.0–3.2 | 3.0–3.7 | 3.0–4.0 | 3.5 |
| Mandible length[2] | 2.1–2.2 | 2.4–2.9 | 2.3–2.7 | 2.5 |
| Orbit diameter[2] | 5.5–8.1 | 4.0–6.3 | 3.6–5.3 | 2.9 |
| Interorbital width[2] | 3.1–3.2 | 3.0–3.9 | 3.2–3.3 | 3.1 |
| Upper-jaw length[2] | 2.5–2.9 | 3.1–3.5 | 2.4–4.0 | – |
| Caudal-peduncle depth[3] | 2.3–2.6 | 2.2–2.7 | 2.3–2.7 | 2.9 |
| Caudal peduncle length[1] | 2.9–4.0 | 4.5–5.2 | 4.9–6.1 | 4.6 |
| Predorsal length[1] | 1.8–1.9 | 1.7–1.8 | 1.7–1.8 | 2 |
| Preanal length[1] | 1.5–1.6 | 1.4–1.5 | 1.4–1.5 | 1.4 |
| Pectoral insertion to pelvic insertion[1] | 4.0–4.5 | 3.4–4.6 | 1.8–2.1 | 3.8 |
| Anal to Caudal length[1] | 3.0–3.1 | 2.9–3.7 | 3.3–3.9 | 3.4 |
| Origin of anal fin to hypural plate[1] | 2.7–3.0 | 3.1–3.3 | 3.0–3.2 | 3.2 |
| Prepelvic length[1] | 1.9–2.2 | 1.8–2.0 | 1.5–1.9 | 1.9 |
| Pectoral-fin length[1] | 3.9–5.7 | 4.8–6.4 | 4.5–6.8 | 5.3 |
| Anal fin height[1] | 4.9–7.4 | 6.1–7.0 | 4.9–4.9 | 5.8 |
| Pelvic-fin height[1] | 5.5 | 5.8–7.9 | 5.7–6.9 | 6.6 |
| Dorsal fin height[1] | 4.4.6.0 | 4.5–6.1 | 4.1–5.8 | 5 |
| Caudal peduncle length/depth | 1.1–1.7 | 1.9–2.4 | 1.9–2.4 | 1.6 |
| Head length/caudal peduncle depth | 1.3–1.5 | 2.8–2.1 | 1.2–1.5 | 1.2 |
| Dorsal rays | I,9 | I,8 | I,8 | I,9 |
| Anal rays | I,9 | I,8 | I,8 | I,9 |
| Pectoral rays | 13–15 | 14–15 | 13–14 | 14 |
| Pelvic rays | I,9 | I,8–9 | I,9 | I,9 |
| Principal caudal rays | 19–23 | 19–22 | 23–25 | 23 |
| Upper procurrent caudal rays | 8 | 7–9 | 7 | 6 |
| Lateral line scales | 89–92 | 73–93 | 59–71 | 92 |
| Scales above lateral line | 21–23 | 18–23 | 16–18 | 25 |
| Scales below lateral line | 14 | 12–16 | 11–13 | 17 |

Note:
Morphometric and meristic analysis of the type series of *G. robusta*, *G. nigra*, *G. intermedia*, and *G. grahamii* for 32 morphological variables presented as a proportion of standard length[1], head length[2], or body depth[3].

*G. robusta* but instead each character was assigned to multiple type series. Only 28% of the time did a morphological character (Tables 1 and 2) align correctly to the types of *G. robusta*, while 63% of the time the characters aligned to *G. nigra* types and 51% to *G. intermedia*, with many of the characters aligning with multiple type series (Table 3). Similar patterns are observed with both *G. nigra* and *G. intermedia*.

**Table 2 Morphometrics and meristics of fresh material in the *G. robusta* complex.**

|  | *G. robusta* | *G. nigra* | *G. intermedia* |
|---|---|---|---|
| Greatest Body depth[1] | 3.9–5.3 | 3.7–5.0 | 3.9–4.8 |
| Head length[1] | 3.4–3.9 | 3.3–3.8 | 3.5–3.8 |
| Head width[3] | 1.5–2.1 | 1.6–2.0 | 1.4–2.1 |
| Head depth[3] | 1.1–1.6 | 1.3–1.6 | 1.2–1.6 |
| Snout length[2] | 3.2–3.8 | 3.1–3.6 | 3.0–4.0 |
| Mandible length[2] | 2.5–2.8 | 2.3–2.8 | 2.4–2.6 |
| Orbit diameter[2] | 3.1–4.9 | 4.4–6.1 | 3.8–6.0 |
| Interorbital width[2] | 3.3–3.9 | 3.1–3.6 | 3.0–3.8 |
| Upper-jaw length[2] | 3.0–3.3 | 1.5–3.4 | 2.5–3.2 |
| Caudal-peduncle depth[3] | 2.1–3.1 | 2.3–2.8 | 2.3–3.2 |
| Caudal peduncle length[1] | 5.0–5.7 | 4.1–4.9 | 4.5–6.1 |
| Predorsal length[1] | 1.8–2.0 | 1.8–2.0 | 1.8–2.0 |
| Preanal length[1] | 1.4–1.6 | 1.4–1.5 | 1.4–1.6 |
| Pectoral insertion to pelvic insertion[1] | 3.6–4.0 | 3.5–4.0 | 3.3–4.0 |
| Anal to Caudal length[1] | 2.9–4.2 | 3.0–3.5 | 3.1–3.8 |
| Origin of anal fin to hypural plate[1] | 2.9–3.6 | 2.9–4.8 | 3.0–4.0 |
| Prepelvic length[1] | 1.8–2.1 | 1.9–3.5 | 1.9–2.0 |
| Caudal-fin length[1] | 3.5–4.2 | 3.9–5.2 | 3.9–4.6 |
| Caudal concavity[2] | 1.8–3.0 | 2.6–3.5 | 2.1–3.0 |
| Pectoral-fin length[1] | 5.3–7.2 | 5.2–6.4 | 5.5–8.5 |
| Anal fin length[1] | 5.5–6.7 | 5.7–6.6 | 5.6–7.8 |
| Pelvic-fin length[1] | 6.7–7.8 | 6.3–7.7 | 6.5–8.8 |
| Dorsal fin length[1] | 4.4–5.4 | 4.5–5.5 | 4.6–6.6 |
| Caudal fin length[1] | 4.0–4.7 | 4.4–6.1 | 4.3–5.2 |
| Caudal peduncle length/depth | 1.6–2.5 | 1.5–2.1 | 1.5–2.6 |
| Head length/caudal peduncle depth | 1.2–1.7 | 1.2–1.7 | 1.2–1.5 |
| Dorsal rays | I,8–9 | I,8–9 | I,8–9 |
| Anal rays | I,8–9 | I,8–9 | I,8–9 |
| Pectoral rays | 14–16 | 14–16 | 13–15 |
| Pelvic rays | I,9 | I,9 | I,9 |
| Principal caudal rays | 19 | 19 | 19 |
| Upper procurrent caudal rays | 8–10 | 6–11 | 7–10 |
| Lateral line scales | 82–95 | 82–89 | 65–87 |
| Scales above lateral line | 23–26 | 20–23 | 17–21 |
| Scales below lateral line | 11–15 | 11–15 | 10–14 |
| Gill rakers | 7–8,2 | 6–7,2 | 7–9,2 |

Note:
   Morphometric and meristic analysis of the fresh samples of *G. robusta*, *G. nigra*, *G. intermedia*, and *G. grahamii* for 36 morphological variables presented as a proportion of standard length[1], head length[2], or body depth[3].

## Molecular analysis

Mapping of the mitochondrial genome resulted in recovery of 14,892 bp of the 16,595 bp reference mtDNA genome. Filtering and haplotyping of the nuclear DNA resulted in a final

**Table 3  Assignment of samples to type material in the *G. robusta* complex.**

| Types | *G. robusta* | | | | | | *G. nigra* | | | | | *G. intermedia* | | | | | | Average |
|---|---|---|---|---|---|---|---|---|---|---|---|---|---|---|---|---|---|---|
| | 105 | 133 | 141 | 145 | 152 | GR1 | 68 | 121 | 132 | 140 | 162 | 100 | 112 | 122 | 125 | 150 | 155 | |
| *G. robusta* | 0.22 | 0.22 | 0.26 | 0.35 | 0.26 | 0.39 | 0.35 | 0.35 | 0.30 | 0.35 | 0.22 | 0.22 | 0.26 | 0.22 | 0.35 | 0.30 | 0.30 | 0.29 |
| *G. nigra* | 0.57 | 0.65 | 0.70 | 0.61 | 0.61 | 0.65 | 0.65 | 0.70 | 0.70 | 0.65 | 0.57 | 0.65 | 0.65 | 0.57 | 0.78 | 0.26 | 0.35 | 0.61 |
| *G. intermedia* | 0.52 | 0.57 | 0.57 | 0.39 | 0.57 | 0.48 | 0.43 | 0.52 | 0.48 | 0.43 | 0.57 | 0.39 | 0.65 | 0.43 | 0.43 | 0.48 | 0.39 | 0.49 |
| Unclassified | 0.17 | 0.26 | 0.13 | 0.22 | 0.17 | 0.09 | 0.13 | 0.09 | 0.17 | 0.22 | 0.30 | 0.22 | 0.13 | 0.26 | 0.09 | 0.13 | 0.39 | 0.19 |

**Note:**
Data for 22 variables for each of the fresh specimens (specimen numbers in bold) were compared to characteristics for specimens in type series (Table 1). Columns provide the proportion of the 22 variables that were consistent with type series data. Fresh specimens that could not be assigned to a given type series are considered as Unclassified. Due to overlap in morphometrics, proportions can add up to greater than 1.

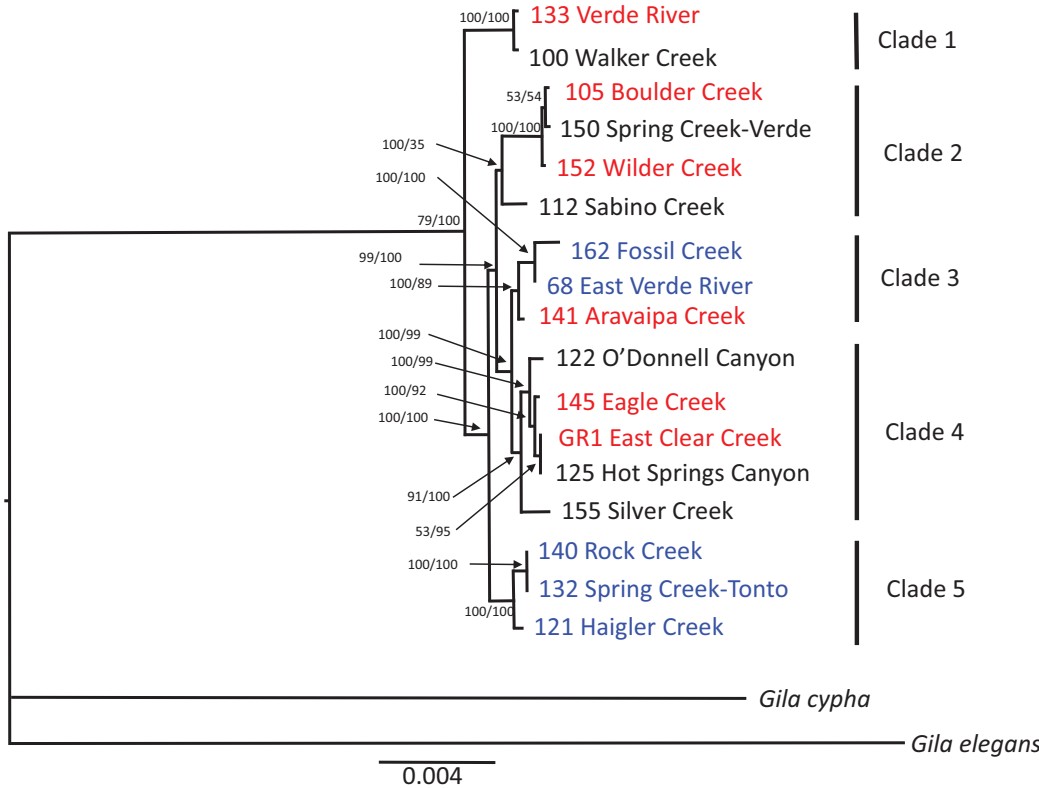

**Figure 2  mtDNA phylogenetic reconstruction.** Phylogenetic tree of mtDNA genome resulting from Bayesian inference for each specimen. Color indicates taxonomic assignment of each sample for *G. robusta* (red), *G. intermedia* (black), and *G. nigra* (blue). Tree rooted with *G. cypha* and *G. elegans*. Node labels are Bayesian probabilities and maximum likelihood bootstraps.

dataset of 1,292 RAD contigs containing 4,821 haplotypes across all individuals, which consisted of 6,658 polymorphisms across 52,483 total bp of nuclear DNA. Analysis of the mitochondrial genome and nuclear datasets revealed high concordance between tree topologies constructed for each of the datasets using both ML and Bayesian tree building methods. Only a single discrepancy was observed between the nuclear dataset and mtDNA dataset: the O'Donnell Canyon specimen is assigned to clade 2 in the nuclear tree and

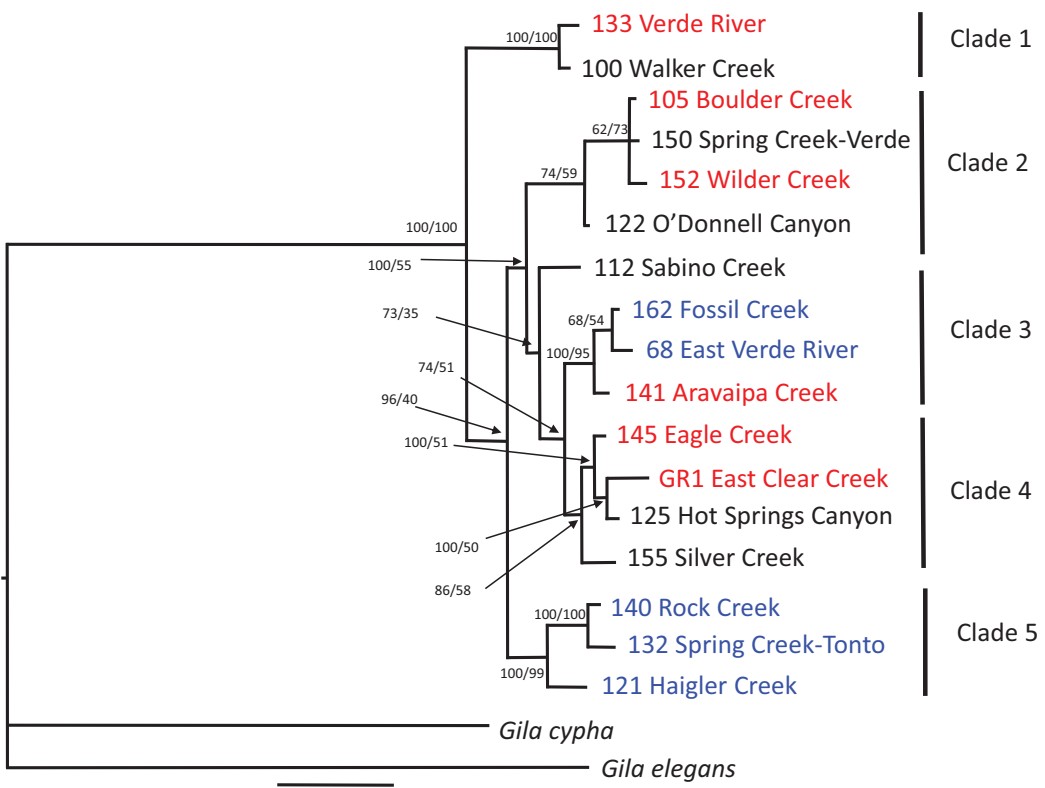

**Figure 3 Nuclear phylogenetic reconstruction.** Phylogenetic tree of nuclear DNA dataset resulting from Bayesian Inference for each specimen. Color indicates taxonomic assignment of each sample for *G. robusta* (red), *G. intermedia* (black), and *G. nigra* (blue). Tree rooted with *G. cypha* and *G. elegans*. Node labels are Bayesian probabilities and maximum likelihood bootstraps.

clade 4 in the mtDNA tree (Figs. 2 and 3), which may be the result of accidental mixing of captive stocks. The results of these molecular analysis are consistent with the morphological finding; *G. intermedia* and *G. nigra* are not distinct evolutionary units and not distinguishable from *G. robusta*. Likewise, the SNAPP coalescent analysis results plotted as a cloudogram reveal high concordance between tree topologies (Fig. 4). Similarly, STRUCTURE (*Pritchard, Stephens & Donnelly, 2000*) analysis and discriminate analysis of principle components (DAPC; *Jombart, 2008*) fail to discriminate the nominal taxa (Fig. S2).

### Morphological analysis of phylogeny

Comparison of morphological characters to the phylogenetic trees revealed nearly 100% overlap of morphological characters between each clade (i.e., there was no differentiation in any morphological character between phylogenetic clades; Table 4) and no diagnostic morphological character was identified that could align with the phylogenetic lineages resolved in this study.

### Test of introgression

The ABBA–BABA test between the *G. robusta* complex and *G. cypha* revealed significant introgression between many of the locations (Table S2) using Patterson's D statistic.

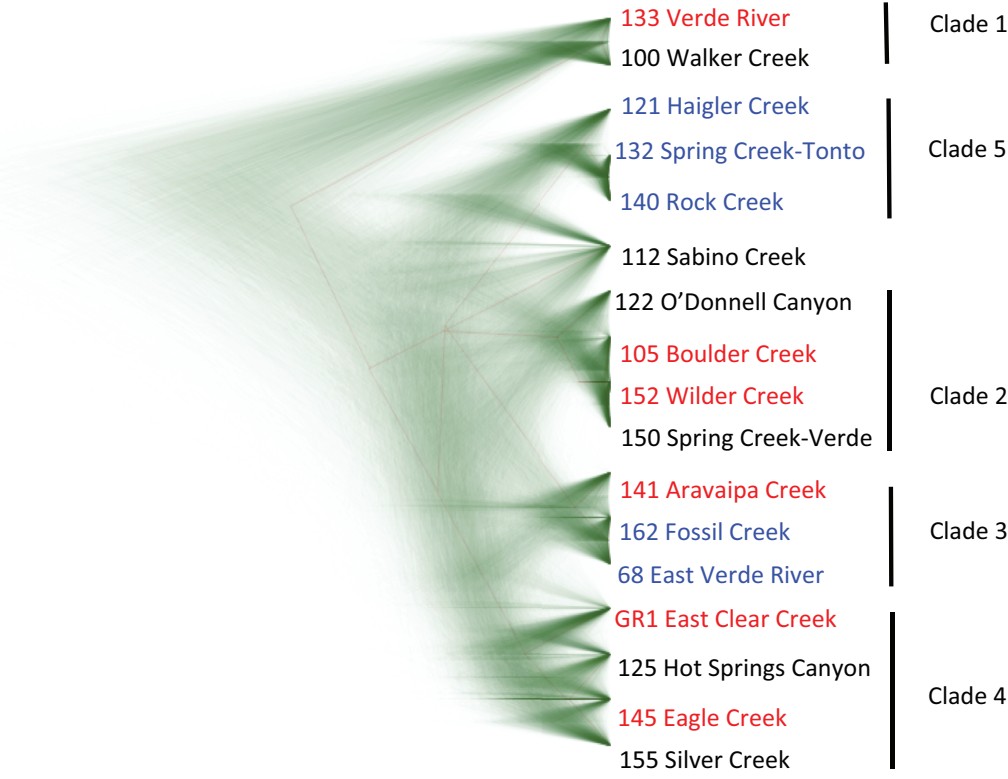

**Figure 4 SNAPP phylogenetic reconstruction in the *G. robusta* complex.** This cloudogram represents the posterior distribution of lineage trees from the Bayesian phylogenetic analysis program SNAPP. Higher density areas indicate greater agreement in tree topologies. Color labels indicate taxonomic assignment of each sample for *G. robusta* (red), *G. intermedia* (black), and *G. nigra* (blue).

However, as expected, the alternative, $F_d$ suggests far fewer locations exhibiting significant introgression. There is evidence for introgression in populations from the Verde River (133), Aravaipa Creek (141), Hot Springs Canyon (125), and Spring Creek-Tonto (132) for both D and $F_d$.

## DISCUSSION

### Taxonomy and nomenclature

*Gila robusta*, *G. intermedia*, and *G. nigra* were originally proposed as distinct species based on the differences observed in the type material, during a time when the natural distribution and variation within this group was unknown. Over the years that followed their description, *G. intermedia* and *G. nigra* encountered a complicated history of synonymy. However, once the synonymy stabilized, the consensus was to treat *G. intermedia* and *G. nigra* within *G. robusta* (see systematic and taxonomic review). This synonymy was widely accepted until *Rinne (1969, 1976)* removed *G. intermedia* from synonymy and *Minckley & DeMarais (2000)* subsequently removed *G. nigra* from synonymy.

We find no evidence, morphological or genetic, to support the current taxonomy. Examination of fresh material revealed that the morphological variability within each of

**Table 4 Morphology by phylogenetic clade in the *G. robusta* complex.**

|  | Clade 1 | Clade 2 | Clade 3 | Clade 4 | Clade 5 |
|---|---|---|---|---|---|
| Body depth[1] | 3.7–5.0 | 3.8–5.2 | 4.2–4.6 | 4.0–5.2 | 3.9–5.3 |
| Head length[1] | 3.3–3.8 | 3.7–3.8 | 3.4–3.6 | 3.4–3.8 | 3.4–3.8 |
| Head width[3] | 1.5–2.0 | 1.6–2.1 | 1.6–1.9 | 1.6–2.1 | 1.4–2.1 |
| Head depth[3] | 1.3–1.6 | 1.3–1.6 | 1.3–1.4 | 1.1–1.4 | 1.2–1.6 |
| Snout length[2] | 3.3–3.6 | 3.3–3.7 | 3.1–3.5 | 3.0–3.8 | 3.1–4.0 |
| Mandible length[2] | 2.3–2.7 | 2.5–2.8 | 2.4–2.8 | 2.5–2.8 | 2.4–2.6 |
| Orbit diameter[2] | 4.7–6.1 | 4–4.1 | 4.4–4.8 | 3.7–4.5 | 3.1–6.0 |
| Interorbital width[2] | 3.1–3.5 | 3.3–3.5 | 3.5–3.6 | 3.3–3.6 | 3.0–3.9 |
| Upper-jaw length[2] | 2.8–3.4 | 3.0–3.3 | 1.5–3.2 | 3.1–3.2 | 2.5–3.3 |
| Caudal-peduncle depth[3] | 2.3–2.8 | 2.2–3.1 | 2.4–2.8 | 2.1–2.8 | 2.3–3.2 |
| Caudal peduncle length[1] | 4.1–5.7 | 4.5–4.9 | 4.5–4.9 | 4.8–5.4 | 4.5–6.1 |
| Predorsal length[1] | 1.8–2.0 | 1.8–2.0 | 1.8–1.9 | 1.8–1.9 | 1.8–2.0 |
| Preanal length[1] | 1.5 | 1.5 | 1.4–1.5 | 1.4–1.6 | 1.4–1.6 |
| Pectoral insertion to pelvic insertion[1] | 3.7–3.9 | 3.7–3.8 | 3.6–4.0 | 3.5–3.9 | 3.3–4.0 |
| Anal to Caudal length[1] | 3.3–3.7 | 3.2–3.4 | 3.0–3.5 | 3.0–3.4 | 2.9–4.2 |
| Origin of anal fin to hypural plate[1] | 3.3–3.6 | 3.1–3.4 | 2.9–4.8 | 3.2–3.6 | 2.9–4.1 |
| Prepelvic length[1] | 1.9–2.1 | 2 | 1.9–3.5 | 1.8–2.0 | 1.9–2.0 |
| Pectoral-fin length[1] | 5.8–6.3 | 4.9–7.2 | 5.2–6.4 | 5.4–6.3 | 5.3–8.5 |
| Anal fin height[1] | 1.7–1.9 | 1.5–1.6 | 1.6–1.9 | 1.5–1.7 | 1.5–2.2 |
| Pelvic-fin height[1] | 2 | 1.7–2.0 | 1.8–2.2 | 1.8–2.0 | 1.9–2.5 |
| Dorsal fin height[1] | 1.5–1.6 | 1.4 | 1.3–1.5 | 1.3 | 1.3–1.9 |
| Caudal peduncle length/depth | 1.5–2.5 | 1.6–2.0 | 1.7–1.9 | 1.8–2.4 | 1.5–2.6 |
| Head length/caudal peduncle depth | 1.2–1.7 | 1.3–1.7 | 1.3–1.5 | 1.2–1.7 | 1.2–1.5 |
| Dorsal rays | I,8 | I,8–9 | I,9 | I,8–9 | I,8–9 |
| Anal rays | I,8–9 | I,8–9 | I,8–9 | I,8–9 | I,8 |
| Pectoral rays | 14–15 | 14–15 | 14–16 | 14–15 | 13–16 |
| Principal caudal rays | 19 | 19 | 19 | 19 | 19 |
| Upper procurrent caudal rays | 7–11 | 9–10 | 6–11 | 8–9 | 7–10 |
| Lateral line scales | 82–86 | 85–87 | 85–89 | 72–82 | 65–95 |
| Scales above lateral line | 20–23 | 21–26 | 21–23 | 18–25 | 17–24 |
| Scales below lateral line | 14–15 | 10–14 | 11–14 | 11–15 | 10–14 |

Note:
Range of morphometrics and meristics by phylogenetic clade for 31 variables. Morphometrics presented as a proportion of standard length[1], head length[2], or body depth[3].

the currently accepted species precluded any distinguishable differences between groups, and individual specimens could not be unambiguously assigned to any type series. The potential characteristic differences observed between each type series, which were originally thought to represent distinct species, reflects the fact that the number of types within a series does not capture the total morphological variation within populations and therefore does not represent what is observed in nature. Different morphological characters assign each individual examined to multiple name-bearing types. This finding is concordant with a robust morphological analysis by *Moran et al. (2017)*, which also could

not resolve these fish into nominal taxa and found that morphometrics and meristics failed to distinguish the three species. Their principal components analysis using geometric morphometrics also could not separate out the three species. After removing outliers and assigning specimens to a priori groups *Moran et al. (2017)* were able to resolve most of the specimens into groups using a canonical variate analysis (CVA). However, the CVA resolved two separate groups of *G. robusta* and appears to be driven by geographic location rather than by evolutionary relationships. The two locations that overlap between *Moran et al. (2017)* and the data presented here show that the *G. robusta* from the Verde River fall out in clade 1 and Aravaipa Creek appear in clade 3 (Figs. 2 and 3) in our data.

We find no evidence to support the validity of *Gila robusta*, *G. intermedia*, and *G. nigra* under any of the more prominent species concepts (i.e., biological species concept, morphological species concept, evolutionary species concept, phenetic species concept, phylogenetic species concept; *Mayr, 1942*; *Simpson, 1961*; *Cronquist, 1978*; *Ridley, 1993*; *De Queiroz, 2007*). Instead, our data appear consistent with phenotypic plasticity for *G. robusta*, and highlight the importance of undertaking a study to test this hypothesis for this species.

It might be argued that introgression within the habitats examined in this study is responsible for the observed morphological and molecular results, but in this regard the International Code of Zoological Nomenclature is clear: "The application of each species-group name is determined by reference to the name-bearing type [Arts. 61, 71–75] of the nominal taxon denoted by the combination in which the species-group name was established" (*International Commission of Zoological Nomenclature, 1999*, Article 45.3). In this case, the morphology of the name-bearing types do not correspond with the forms to which the names have been applied in natural populations. Likewise genetic approaches (both STRUCTURE and DAPC) fail to discriminate among the nominal species groups (Fig. S2) and these nominal species do not resolve as monophyletic in the phylogenetic analyses (Figs. 2–4). Because there are no diagnostic morphological or molecular characters that consistently distinguish the populations to which the names have been applied, the species names should not be applied. The data presented here indicate either a single morphologically plastic species, extant populations that consist almost entirely of hybrid individuals of mixed ancestral lineages, or a combination of both. Thus, based on the inability to unambiguously assign individuals to a single taxonomic category, our corresponding conclusion is to synonymize *G. nigra* and *G. intermedia* with *G. robusta* (the name with priority). This conclusion is reinforced by our findings that the taxonomic key and underlying data used to distinguish *G. intermedia* and *G. nigra* from *G. robusta* fail to assign even the type specimens unambiguously to a single species. Using mean differences between populations to justify species distinction subsumes the extent of natural variation within conspecific populations, but also, (as is the case here) can lead to polyphyly within nominal species.

## Patterns and drivers of variation

*Gila intermedia* and *G. nigra* were regarded as distinct species based on mean differences between populations inhabiting different streams (*Rinne, 1969*, *1976*; *Minckley &*

*DeMarais, 2000*). Some authors suggest that these patterns are based on environmental differences such as water depth and speed (*Miller, 1946*; but see *Rinne, 1976*). Plasticity resulting in a gradation of characters by stream size and current is observed in many species of freshwater fishes (*Hubbs, 1940*). For example, the bluehead sucker, *Catostomus discobolus*, varies morphologically according to size and flow of the water it inhabits (*Sigler & Sigler, 1996*). Similarly, the blacktail shiner, *Cyprinella venusta*, exhibits morphological variation between steams and reservoirs and the magnitude of morphological change is correlated with size of the reservoir (*Haas, Blum & Heins, 2010*). The brook charr, *Salvelinus fontinalis*, exhibits variation in caudal fin size and body shape with water velocity (*Imre, McLaughlin & Noakes, 2002*). In each of these cases, the population mean differences in morphology are responses to environmental conditions indicating morphological plasticity rather than diagnostic evolutionary traits that could define species.

The morphological and molecular patterns observed across the geographic range of these nominal species have prompted a number of hypotheses. First, present-day taxa may be relics from the last pluvial period when the wetter climate resulted in higher connectivity, with subsequent post-glaciation aridity resulting in local divergence via selection and genetic drift (*Williams et al., 1985*; *Meffe & Vrijenhoek, 1988*). In this case, selection would drive a phenotypic response in current taxa irrespective of evolutionary history, while drift should result in a geographic component to the phylogeny. Alternatively, perhaps current taxa were once isolated and are now hybridizing, yielding the morphological variation observed today (*Gerber, Tibbets & Dowling, 2001*; *Osborne et al., 2016*; *DiBattista et al., 2016*). Finally, current taxa may be the result of ancient admixture and subsequent isolation resulting in conflicting morphological and molecular signals for distinguishing species. It is difficult to separate these alternate hypotheses (*Eaton & Ree, 2013*; *Merrill et al., 2015*; *Eaton et al., 2015*; *Martin et al., 2015*) but in the case of *G. robusta* it is likely that a combination of these hypotheses is responsible for the observed patterns. Additional research that focuses on testing each of these alternative hypotheses in a robust way would be necessary to fully understand the mechanisms responsible for the morphological variation observed in this species.

## Phylogenomics and hybridization

With the exception of clade 5, no lineage recovered from the phylogenetic analyses (Fig. 2) consists of a single species as currently defined. However, the fact that clade 5 consists exclusively of *G. nigra* may be an artifact of the small geographic area and proximity of sampling locations (Fig. 1). Individuals identified as *G. nigra* also occur in clade 3 with *G. robusta*, so while it is possible clade 5 represents a geographically restricted lineage, additional geographical sampling will similarly likely erode the unity of this clade. In any case, whatever the finding with clade 5, our data are consistent with previous studies based on allozyme, mtDNA, and microsatellite markers that likewise failed to find diagnostic characters among these nominal species (*DeMarais, 1992*; *DeMarais et al., 1992*; *Dowling & DeMarais, 1993*; *Dowling et al., 2015*; *Marsh, Clarkson & Dowling, 2017*).

The average most recent common ancestor of each of the clades resolved in this study is 63 kya (51–76 kya 95% HPD interval) with the most recent common ancestor of all

populations 119 kya (97–140 kya 95% HPD interval). The divergence times of this group fall well within the last glacial cycle and with such recent divergence of the populations, it seems likely to be linked to post glacial warming and subsequent transitions from the wetter climate of the Last Glacial Maximum to the more arid climate of today (*Williams et al., 1985*; *Meffe & Vrijenhoek, 1988*).

Tests of genomic admixture (hybridization) indicate that *G. cypha* historically interbred with all three nominal species of the *G. robusta* complex, or interbred with the recent common ancestor of each of these populations. Despite the fact that *G. cypha* and *G. elegans* are currently highly endangered, with ranges restricted to the main stem of the Colorado River, evidence suggests that at one time their ranges may have overlapped. The type locality of both *G. robusta* and *G. elegans* is the Zuni River, New Mexico, and types were collected on the same expedition at the same locality. Unfortunately, due to the nature of the ABBA–BABA test, we were unable to test for significant introgression between *G. elegans* and *G. robusta*.

It is possible that introgression may have resulted in phenotypic traits passed from *G. cypha* to *G. robusta*, contributing to the morphological variation observed here. However, no study assessing the heritability of phenotypes between these species has been conducted to test this hypothesis. Similarly, no test of morphological plasticity with regard to stream condition has been conducted. These deficiencies need to be addressed in order to fully understand the patterns observed within natural populations of this species.

## Management implications

Our data do not support the current taxonomy of *Gila robusta*, *G. intermedia*, and *G. nigra*. Instead, we find evidence that may correspond to environmental condition and geography more than currently accepted taxonomy. Given the propensity of the cyprinids for introgression (*Briolay et al., 1998*; *Rosenfeld & Wilkinson, 1989*; *DeMarais et al., 1992*; *Dowling & DeMarais, 1993*; *Gerber, Tibbets & Dowling, 2001*), speciation within the Lower Colorado River Basin seems unlikely. Hybridization in fishes is a common occurrence (*Allendorf & Leary, 1988*). About 30% of known hybrids in freshwater fish species belong to the Cyprinidae, with ongoing intergeneric hybridization continuing between species that diverged 10–15 million years ago (*Briolay et al., 1998*). The *G. robusta* species complex is no different; hybridization producing viable offspring is a common occurrence (*Gerber, Tibbets & Dowling, 2001*; *Marsh, Clarkson & Dowling, 2017*). This history, coupled with the cyclical nature of glacially driven pluvial periods makes it unlikely that evolutionary forces will induce speciation in the absence of permanent barriers to gene flow. Instead, these populations have likely experienced repeated cycles of isolation during dryer periods, followed by connectivity during wetter periods. At each phase, selective forces could favor different phenotypes in different isolated regions, but without reproductive isolation, these lineages could mix during each cycle. These fluctuating selection regimes combined with introgression could result in the complex array of morphological variation observed within this species. In these circumstances, management should focus on maintaining genetic diversity to ensure long-term persistence. Greater genetic diversity is associated with enhanced mean fitness (*Quattro & Vrijenhoek*,

*1989*; *Reed & Frankham, 2003*) and decreased extinction risk (*Frankham, 2005a*; *Evans & Sheldon, 2008*), so management should focus on preventing the loss of genetic diversity upon which long-term persistence and adaptability depend (*Vrijenhoek, Douglas & Meffe, 1985*; *Quattro & Vrijenhoek, 1989*; *Frankham, 2005b*; *Hancock et al., 2011*).

Why have the nominal *G. intermedia* and *G. nigra* persisted as taxonomic entities in the face of much evidence to the contrary? In addition to legitimate uncertainties about evolutionary partitions, these may be examples of geopolitical species (sensu *Karl & Bowen, 1999*), with species defined by geography, and taxonomic status subsequently maintained to support conservation priorities. There is currently a heated debate in the literature between conservationists and taxonomists regarding the need for fixed taxonomic entities on which to apply conservation priorities versus the dynamic nature of taxonomy that allows for revision of taxonomic hypotheses with new data, methodology, or insights (*Garnett & Christidis, 2017*; *Thomson et al., 2018*). This situation is likely exacerbated by legislation that emphasizes protections of fixed species, such as the Endangered Species Act, but does not allow for taxonomic revision and advancement. For instance, during heated debate over the taxonomy of the endangered green sea turtle (*Chelonia mydas*), *Karl & Bowen (1999*; see also *Bowen & Karl, 1999)* observed that there are scientifically sound reasons for conservation of isolated populations (which is also true within *G. robusta* in this case), but nominal taxonomy is not one of them. Conservation priorities may change over time to allow adaptive management, but taxonomy should only be influenced by scientific data as applied through the rules of the International Commission of Zoological Nomenclature. As with the green sea turtle, the conservation status of these fishes remains a separate issue, but taxonomic assignments that do not meet the standards of the Code should be put aside to allow researchers to reassess the true relationships within the *Gila* of the Lower Colorado River Basin. Range-wide genetic surveys should be undertaken to fully identify genetically distinct units, and their geographic extent, in conjunction with thorough morphological analyses to determine species boundaries in this system, which will be facilitated by first purging the incorrectly assigned nomenclature and starting with a clean slate.

## CONCLUSIONS

The results of this study question the validity of the taxonomic names, but do not indicate that protections for this species should cease. Instead our results indicate the need for protection at a population level, to maintain genetic diversity and morphological variation, rather than three nominal species for which no diagnostic morphological or genetic characters exist. The Endangered Species Act defines a species to include "any subspecies of fish or wildlife or plants, and any distinct population segment of any species of vertebrate fish and wildlife which interbreeds when mature" (Section 3(15), ESA 1973, 1978). Neither our data, nor that of previous studies, indicate a single well-mixed population across the Lower Colorado River Basin. In fact, based on previous work, genetic structure may exist among many of the populations analyzed, but this population structure is not aligned with the three currently recognized taxonomic units; the shallow divergences between samples is indicative of population differences, but no species

level genetic divergence has been observed. Our study was designed to examine a phylogenetic question and sampling was not sufficient for addressing population level questions. Thus, while these data indicate such population genetic studies are warranted, we caution against inferring population level conclusions until such robust surveys are completed. We recommend that this species be managed as distinct population segments until the additional studies outlined herein are completed.

## ACKNOWLEDGEMENTS

We thank Richard Pyle of the Bishop Museum for assistance with taxonomy and nomenclature as well as his knowledge of the ICZN Code. We also thank Matthew O'Neill, Clay Crowder, and Julie Carter of the Arizona Game and Fish Department; Cassie Ka'apu-Lyons, Emily Conklin, Mykle Hoban, Ingrid Knapp, Mahdi Belcaid, Anne Lee and the ToBo Lab members at the Hawaii Institute of Marine Biology; Jon Puritz at University of Rhode Island for his help with dDocent; Anthony Montgomery at the USFW; and Sandra Raredon and Jeff Williams of the National Museum for their assistance with the type material. Thanks to editors James Reimer, Tomas Hrbek and two anonymous reviewers for comments and suggestions that greatly improved the manuscript. This is University of Hawaii School of Ocean and Earth Science and Technology contribution 10451 and Hawaii Institute of Marine Biology contribution 1737.

### Funding

This work was funded by NSF OCE 12-60169, OCE-15-58852, the Arizona Game and Fish Department and the Seaver Institute. There are no additional external funding sources for this project. The funders had no role in study design, data collection and analysis, decision to publish, or preparation of the manuscript.

### Grant Disclosures

The following grant information was disclosed by the authors:
NSF OCE 12-60169, OCE-15-58852, the Arizona Game and Fish Department and the Seaver Institute.

### Competing Interests

Robert J. Toonen is an Academic Editor for PeerJ.

### Author Contributions

- Joshua M. Copus conceived and designed the experiments, performed the experiments, analyzed the data, contributed reagents/materials/analysis tools, prepared figures and/or tables, authored or reviewed drafts of the paper, approved the final draft.
- W.L. Montgomery performed the experiments, authored or reviewed drafts of the paper, approved the final draft.
- Zac H. Forsman analyzed the data, authored or reviewed drafts of the paper, approved the final draft.

- Brian W. Bowen contributed reagents/materials/analysis tools, authored or reviewed drafts of the paper, approved the final draft.
- Robert J. Toonen conceived and designed the experiments, contributed reagents/materials/analysis tools, authored or reviewed drafts of the paper, approved the final draft.

## DNA Deposition

The following information was supplied regarding the deposition of DNA sequences:
GenBank SAMN07627206–SAMN07627224.

## Data Availability

The raw data are provided in the Supplemental Files.

## Supplemental Information

Supplemental information for this article can be found online at http://dx.doi.org/10.7717/peerj.5605#supplemental-information.

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
