# Peer review of "Geopolitical species revisited: genomic and morphological data indicate that the roundtail chub Gila robusta species complex (Teleostei, Cyprinidae) is a single species"

_PeerJ, doi:10.7717/peerj.5605_

## Round 0.1 · original submission · Major Revisions

I have heard back from two reviewers, who were generally very supportive of your work, but also offered many constructive comments. While none of these comments alone would require major revision, taken together, some extensive rewriting may be needed, and hence my decision is 'major revisions' are needed.

I look forward to seeing your revised paper.

Reviewer 1 ·

Basic reporting

The manuscript is written in clear, unambiguous, professional English throughout.

The introduction & background show context. The context is the extraordinary confusion created by previous inexpert and inadequate species descriptions and taxonomic revisions. The authors are admirably restrained in their criticism of this taxonomic confusion.

I’m not an expert on these fishes, so I wouldn’t be able to tell if the referencing was inadequate or missing key citations to important previous work. The referencing throughout (especially in the taxonomic summary) seems to be comprehensive.

The taxonomic revisions by Rinne seem to be a really important source of the extensive taxonomic confusion surrounding species of Gila. So I was a little distressed to find that none of those papers (Rinne 1969; Rinne 1976; Rinne & Minckley 1970) is in the References list. From mousing around I think Rinne 1969 is his Masters thesis at ASU; and Rinne 1976 is his article in the Wasmann Journal of Biology? Also there is a typo on line 825: “Tosada” should be “Posada”.

The structure conforms to PeerJ standards and norms for this type of phylogenetic taxonomic study.

The figures are relevant, clear, and informative. There are some errors in the figures. The legend for Fig. 1 says “stars represent locality of type series for G. robusta (red) G. intermedia (black) and G. nigra (blue)” but there are no stars; I think the authors mean the squares? The type locality for G. robusta is outside of the map area and is not shown on a specific drainage, which seemed confusing. Also the site number 122 (the collection location for the individual called “122 O’Donnell Canyon” in Fig. 2) is not shown in Fig. 1. Similarly, the collection locations for the G. cypha and G. elegans individuals is not shown in Fig. 1 or described in the text.

Raw data were supplied.

Experimental design

The paper is based on original primary research that falls within the scope of the journal.

The research question is well defined, relevant & meaningful. It includes a clear statement on how the research fills an identified knowledge gap.

The study is based on rigorous investigation performed to a high technical & ethical standard. This type of study is exactly what RADseq methods were developed for: the volume and quality of data generated by this method for this study gives the authors a lot of power to test the three-species hypothesis as well as to explore patterns within those data that might be more consistent with other hypotheses. The data in this study are state-of-the-art.

The methods are described with sufficient detail & information to replicate. However, I did not understand how the authors gave provisional species names to the newly collected individuals that were the material for the morphological and genetic data collection. The text (lines 224-227) says that “Due to the problems associated with the taxonomic key (Moran et al. 2017) the current practice of species identification for managers and researchers working with the G. robusta complex within the Lower Colorado River Basin is based on drainage location. We follow this convention because no alternative method of assignment is currently available.” It makes sense that this is the current practice. But it was hard to understand from the map in Fig. 1 how this worked for the authors in this study. For example, some of the red locations for G. robusta (labelled 141 and 145 in Fig. 1) come from the southern drainage that includes the G. nigra type locality (the blue square) as well as one of the G. intermedia locations (125). Also the type locality for G. robusta (the red square in Fig. 1) is off the map and not shown on a specific drainage. Maybe this means I just don’t understand what ‘drainage’ refers to: is each of the light blue lines in Fig. 1 a single drainage? Or do collections of light blue lines leading to one of the dark blue lines form a single drainage? I needed a more complete and clear explanation of this working definition of how to apply the three provisional species names to individual fish. I realize that this is in essence the problem the authors are trying to sort out, but the starting point for the reader’s understanding of the study is an understanding of how one of the three provisional species names was applied to each individual fish that was collected for the genetics and morphological parts of the study. I didn’t understand how that was done. Possibly what I needed was an additional figure that shows the geographic ranges of the three nominal species mapped onto that same illustration of river drainages in Fig. 1, maybe with the individual creeks and larger tributaries colored red, black, or blue to show which ones harbor G. robusta, G. intermedia, or G. nigra.

Validity of the findings

The data are robust, statistically sound, & well-controlled. I had two questions about the data analysis. These are also my only substantive criticisms of the study and interpretation.

First, if the individual fish sampled for this study all come from a single biological species (as the authors propose), then recombination among many of the contigs from the ezRAD data should have scrambled their genealogical history. The authors argue (lines 552-553) that they have found “evidence for five distinct clades that may correspond to geography more than currently accepted taxonomy.” The authors claim in the Discussion (lines 596-599) that “Our data do not indicate a single well-mixed population across the Lower Colorado River Basin. In fact, clear genetic structure exists among many of the populations analyzed (especially between the five well-resolved clades), but this population structure is not aligned with the three currently recognized taxonomic units.” Because recombination obscures genealogical relationships, I think these statements are far too strong and are not supported by the data.

For samples of many unlinked loci from across the genome (such as ezRAD contigs), analyzing all those unlinked loci by concatenating the sequences together (as if they all came from the same single sex chromosome or some other single nonrecombining unit or haplotype) within each individual does not give a very meaningful illustration of the history of those individuals or their genealogical relationships to each other. Instead, each of the ezRAD contigs has its own genealogical history that is independent of the other contigs because of recombination and independent assortment. This is the problem with phylogenetic thinking in a population context for sexual organisms. For example, imagine that individuals 100 and 150 swam downstream, met, and mated in the main tributary that they share (the dark blue line to the lower left of those two labeled black points in Fig. 1). Their offspring would have half of the contigs from 100 and half from 150, and it seems unlikely that a phylogenetic analysis of those two sets of mixed ezRAD contigs (like the analysis presented in the manuscript) from those offspring would correctly assign those offspring to either clade 1 or clade 2, because recombination within the two parents would scramble the apparent signal of phylogenetic relationships among those offspring relative to the parents and relative to the other 15 sampled individuals.

So I don’t agree with the authors that these five clades are real entities, that they are possibly about to evolve into new species, or that they represent promising targets for conservation efforts. The five clades seem likely to be artifacts of concatenating together the phylogenetic signal from hundreds of ezRAD contigs that in nature are unlinked due to recombination and independent assortment (and have no real phylogenetic signal when concatenated together as if they were linked). Unless I have badly misinterpreted the methods, this seems like a really surprising error for the authors to have made (especially the senior authors who are experienced and skilled population geneticists). So I apologize if my criticism is based on a misunderstanding.

I think a coalescent analysis (instead of a phylogenetic analysis) that uses each of the gene trees from all of the (presumably unlinked) contigs seems like a better analysis. Could the authors consider doing that population genetic (rather than phylogenetic) analysis? It would be a nice complement to the clustering analysis proposed in my next comment below. Or, again, possibly this is an analysis for a different paper. Either way, I think at the very least the authors need to address the problem of recombination among unlinked ezRAD contigs and explain or account for how such recombination might influence their interpretation of a phylogeny of individuals in which unlinked parts of the genome are treated as if they were all linked in one long haplotype. I think that phylogenetic analysis is wrong (based on concatenated haplotypes of unlinked loci), so again at the very least the authors need to convince me (and other readers) that this phylogenetic analysis is relevant and informative.

One possible approach would be to treat the three sets of samples of the nominal species as possibly interbreeding populations, and fit isolation-with-migration models to the multilocus ezRAD contig alignments (using each contig as a locus). The authors could use those models to ask whether the population divergence times among robusta, intermedia, and nigra are detectably different from zero (that is, whether there really are three distinct populations or species that have been evolving independently); they could include the mtDNA haplotypes as an additional locus, one with a reasonably well-known mutation rate, and use that mutation rate calibration to convert the model parameters to population units (especially to convert the divergence time estimates to years). That kind of analysis could be done using IMa2, or (maybe even better) using one of the ABC methods.

The authors argue (lines 599-600) that “Our study was designed to examine a phylogenetic question and sampling was not sufficient for addressing population level questions”, but intensive sampling of individuals is not required for IMa model-fitting. Those models depend mainly on deep sampling of genomes (loci), which ezRAD does very well. I think the authors’ data are well suited to an isolation-with-migration analysis.

Second, the analysis of the genetic data seems to show that no clades consisted solely of individuals from one of the three ‘species’. On that basis the authors conclude that all of those individuals should be assigned to one species name. That may be an appropriate conclusion, but that analysis only shows that none of the three ‘species’ forms a monophyletic group; the analysis doesn’t consider whether there are multiple genetically distinctive clusters of genotypes within those data. Asking whether there are distinctive multilocus genotype clusters is not the same as asking whether there are long branches in the concatenated gene tree (Fig. 2). The authors argue that these five distinctive clades in Fig. 2 might be important evolutionarily distinct units, but I argued above that this phylogenetic analysis seems flawed.

As I read the description of the data collection I was sure the authors would eventually describe how they carried out a STRUCTURE analysis or some other multilocus clustering analysis that would help them decide whether the fish they sampled cluster into one or more than one distinctive group of multilocus genotypes. But that analysis isn’t in this manuscript. If that’s an analysis for another paper, then that would be ok. But it would be great if the editor wants to encourage the authors to add that population genetic analysis to the phylogenetic analysis presented in the submitted manuscript. That clustering analysis would be a nice complement to the isolation-with-migration model fitting that I suggested above. Those analyses together would go a long way toward testing the authors’ assertions that (1) the three named species taxa are not monophyletic clades AND (2) all individuals belong to one cluster of genotypes that represents a single biological species.

At this point in writing the review, I wanted to look at some of the recent molecular studies that have previously tried to sort out this problem, including Schonhuth et al. 2012, 2013; Dowling et al. 2015; and Marsh et al. 2017. But again none of those four articles is in the References list. It seems important, especially in an article like this where the goal is to sort out problems in the previously published literature, for the authors to carefully check all of the literature citations in the text and match them to all of the items in the References list.

The conclusions are well stated, linked to the original research question, and limited to supporting results. I thought there might be one feature of both the review (of the taxonomic history) and the interpretation of the phylogenomic patterns that could be expanded on. The authors note in several parts of the text that the genetic data are not consistent with phylogenetic patterns expected to be produced by processes acting on several reproductively isolated species. And in several places in the text the authors note the possible contribution of hybridization to account for some of the observed genetic and morphological variation. But the authors don’t say anything specific about knowledge of actual reproductive isolation (or reproductive compatibility) among individual fish from different populations for any of the three named species taxa (robusta, intermedia, nigra). Maybe the reference to heritability of phenotypic differences (lines 544-545) is a reference to reproductive isolation and hybridization? Is nothing known about whether males and females from these different locations or ‘taxa’ can mate with each other? If (like some other fishes) gametes can be artificially stripped from adults, can ‘hybrid’ offspring be produced by artificial insemination? Are fertilization rates or other measures of gametic compatibility lower for ‘hybrid’ crosses than for ‘conspecific’ crosses? What do those ‘hybrids’ look like? Are they viable? This seemed like an obvious important gap in knowledge: if the gap is real, the authors should be encouraged to highlight it prominently. If the gap is only apparent, the authors should be encouraged to fill it in by reviewing previous studies of gamete compatibility, mating behavior, viability and morphology of hybrids, or other direct evidence for reproductive isolation.

Additional comments

The lengthy quotation from the note appended to USNM 246 (lines 98-101) is remarkable. If it’s possible, could the authors consider adding to the manuscript a figure that shows a photograph of that note? Maybe that’s not very valuable as data, but as an interested reader I would have loved to see the actual 1945 note itself in R. R. Miller’s handwriting as a sort of historical artefact. The subsequent paragraph (lines 103-119) is again remarkable: taxonomy is hard enough, but the extent of taxonomic confusion about these fish that was created by the biologists (rather than by the biology) seems extraordinary.

As an aside, I’ve worked with some museum collections but never with an accession number so low. These are old, important, original fish collections!

Lines 58-62: The paragraph seems out of place: it refers to hypotheses used to account for patterns, but the patterns haven’t been described yet. The reader doesn’t know what ‘patterns’ are until after reading the section ‘Systematic and taxonomic review’ (which is excellent). Maybe instead of “invoked to explain the observed patterns” it might help to say specifically “invoked to account for geographic variation in morphological and ecological traits within and among species of Gila.”

Lines 79-80: “in part due to the reliance of population means as diagnostic characters and variance among individuals within and among locations.” I think the authors mean “in part due to the reliance on populations means as diagnostic characters, and in part due to confusion arising from variation [not variance] among individuals within and among locations.”

Lines 401-403: “For example… only 28% of the 32 morphological characters examined (Table 1,2) aligned with the types of G. robusta, but instead 63% aligned to G. nigra and 51% to G. intermedia (Table 3).” That seems to add up to 142%, so I think the sentence needs to be rewritten or maybe broken up into two sentences that correspond to two different variables or quantities (I think that’s the source of the confusion but I can’t tell from the context of the sentence).

Part of the text (lines 396-408) seems to be in gray rather than black font?

Line 433: Spell out Gila to avoid starting the sentence with an abbr.

Line 451: “name-bearing”, not “name bearing”.

Line 475: “let alone” is too colloquial here.

Lines 501-516: “current taxa” is potentially confusing because the previous paragraph discussed topics related to the speed or velocity of water flow (i.e., water currents); maybe use “present-day taxa” instead? Similarly, “drift” has a hydrological as well as a genetic meaning; maybe use “genetic drift”?

Line 543: Indent for a new paragraph here?

Line 582: In this sentence “codified” and “Code” are redundant: the C in ICZN stands for Code. Instead how about “as applied through the rules prescribed by the ICZN.”

Lines 573-588: This discussion of the idea of geopolitical species was interesting and useful. It seemed to me while reading this paragraph that part of the problem is specific to the US Endangered Species Act and its emphasis on conservation at the level of taxonomic species. Is this a universal problem, or is it more relevant to conservation problems and priorities in the USA (I don’t know because I’m not a conservation biologist and not an American)? Maybe the authors could highlight this problem as part of the explanation for motivated reasoning on the part of taxonomic splitters?

Lines 589-590: “To be perfectly clear” is too colloquial.

Reviewer 2 ·

Basic reporting

Nicely written, bogs down in the details of the taxonomy, some of which might be better served in references to other treatments of similar scope.

I pointed out a couple of issues with citations.

Rather than colors, could you include the 'putative' species designation on the phylogeny to show polyphyly more readily?

Experimental design

see comments below

Validity of the findings

Im not so sure that these data add much to what has already been done, but it is nonetheless a cohesive body of work that includes a variety of molecular and morphological approaches. To that end, the authors are quite convincing that robusta should be recognized but other species subsumed.

Additional comments

Copus et al.

Geopolitical species revisited: Genomic and morphological data indicate that the Roundtail Chub Gila robusta species complex (Teleostei, Cyprinidae) is a single species

Minor edits and Queries:

L27: (~ 90% complete) - not sure if this is useful here

L48: citation or description (assume both?)

L58-62: This is an awkward paragraph, that contains too little information to stand as on its own. Suggest appending to following paragraph.

L101: Not getting how this relates to taxonomic confusion unless youre implying that J&E thought these were cods. Appear to be more clerical errors.

L109: from the Zuni River

L122: G. intermedia has a complex taxonomic history (this isn’t necessary here)

L151: gets a little confusing, maybe add a reference for ‘original description’

L154: Why not sufficient?

L224: This should be somewhere in the introduction with an explanation of why individual characters are not useful

L248: might indicate in parentheses which were standardized by what (since the table is supplemental)

Do any of these morphological measurements bear on the aforementioned “the inferior fins especially are quite small”

L255: greater than 10Kb?

L265: manufacturers’

L267: to 1μg of template DNA (something seems missing here)

L284: curious, are there any potential issues with orthology/paralogy?

L286: paired

L306: done by position or just wholescale by locus? Why not retain the mixed model of mtDNA and nDNA in the phylogenetic analyses?

L378: Rather than looking for individual characters, could you represent this in space via a PCA, or is the number of specimens versus the number of characters an issue?

L427: I am confused by what is meant here, what exactly are you comparing? I think, by the table, you are calculating means or comparing ranges by molecular clade, correct?

L442: first mention of reproductive isolation, how measured? I’m guessing this was simply inferred?

L443: this is interesting, is the inability to discriminate limited to fresh materials (with wider geographical distribution), or does this extend to both data sets. Any evidence that this issue is more applicable to the fresh versus the archival specimens?

L447: Answers comment above, but note that this should be reinforced in the results where there is little attention to the fact that such differences in fact might exist in the types (unless you are arguing the converse, but I didn’t get that impression)

L454: see comment L378 above, I believe this would be a more robust analysis than unitary characters, at least as a means to visualize ‘morphospace’

L475: see L443 comment above. Some of this discussion is a bit disjunct, and might be placed together for a better reading mss

L460: Is there any chance that more recent hybridization might have obscured whatever might be seen in the types? That is, are the types just as confusing as the fresh material?

L495: Important to detail which concepts you’re talking about, unless of course, you’re implying that the current data wouldn’t support any commonly held concepts

L497: Kind of a strawman without presenting the basis for how cryptic speciation was actually diagnosed by Marsh

L526: Marsh et al. is not in citations

L527: with all of the available information, would coalescent type approaches be useful?

L599: This is a salient point, suggesting how this manuscript is predominately molecular in scope, exhaustive and important, but most of the manuscript focuses on the morphology.

L605: Would be best to include this in the introduction somewhere.

L610: Do other of the more recent treatments (Marsh in particular) bear on this as well – is that manuscript perhaps better with more comprehensive morphological sampling? I think a lot of this belongs in the introduction, no matter how awkward that might be to write

L768: citation repeated (should be Marsh 2017?)


Overall: My impression of this manuscript are favorable to a large extent, but there exists some confusion over what has been done already (new morphological analyses by March, all of the pre-existing molecular data) and what exactly this manuscript adds to the question of species validity. I think the manuscript could do with a tightening of the Introduction, touching on what has been done historically (and especially more recently, e.g., Marsh and Page), and what value the huge amount of molecular data adds to the discussion. The authors are quite convincing that fresh materials do not support any species boundaries as previously defined, at least morphologically (although sample sizes are very small), but it seems from reading the discussion that this is a foregone conclusion from what has been done recently. What is new here, and I believe might be the better focus, is the large amount of concordant nuclear and mitochondrial DNA data and the phylogenetic analysis that it affords.

---

## Round 0.2 · Minor Revisions

Dear Authors,

First, I have taken up editing your paper after the second round of reviewer evaluations. So I wanted to read through the previous version to get a better handle on your paper.

I agree with both reviewers that the paper is in good shape and merits publication. However, although both referees recommend publication, I would like you to address and explore a couple of more issues before I recommend publication as well. Essentially I think you can make a much stronger case of having just one species using the molecular data you have at hand.

1. I appreciate that you did Structure analysis, and same as reviewer 1 I suggest that you report these results. Besides the potential issues of overparametrization pointed out by reviewer 1, the problem is more fundamental. If you use the Evanno method to identify biological groups, the method will not identify less that two groups (you can never conclude that you only have one biological group). This is consequence of comparing second and third order differences. So look at the likelihoods of one group and two groups; it is quite likely that the likelihood of having one group will be greater than that of two groups (or the difference between one and two groups will not be appreciable). Finally, the Structure pattern that you are seeing (each individual has equal probability of belonging to one or the other group) is a classic sign of these two groups not existing.
2. I suggest that you run PCA or DAPC analysis. Since DAPC models the data to discriminate between within and between group differences, maximizing between group differences, if you are unable to discriminate between these clusters then you really do not have different groups. You can also produce visually pretty graphics in DAPC to show this.
3. It is unclear to me how you decided what is a clade (why did you partition your phylogenies into clades shown in 2-4?; eg. why are clades 3 & 4 two clades and not one?; eg. why was sample 112 not labeled as a clade, albeit with only one individual?; was any explicit criterion used in delimiting clades?), and what does this designation of clades add to your study. To me, what is important and again will support your conclusion that you do not have different species, is the fact that your “species” are not monophyletic. This is what should be emphasized on the figures.

Otherwise I think you have a solid paper that tackles a potentially sensitive issue head on. And although I would rather not return the MS back to you, addressing these issues and including these analyses and results will only make your case stronger.

Sincerely,

Tomas Hrbek

Reviewer 1 ·

Basic reporting

I'm satisfied with the authors' thorough and positive response to my previous review and comments.

Experimental design

I'm satisfied with the authors' thorough and positive response to my previous review and comments.

Validity of the findings

I'm satisfied with the authors' thorough and positive response to my previous review and comments.

Additional comments

Thanks for a very thorough revision. This is the kind of interaction with authors that makes peer review worthwhile for me.

I had just one additional comment (not a criticism). The authors included an example of their STRUCTURE results in the cover letter, and argued (convincingly) that their sample of individual fish is too small to justify publishing this analysis. I agree. But I wanted to comment on the quality of that result. Although I'm not an expert on this type of clustering analysis, I think these STRUCTURE results are actually consistent with all of the authors' other phylogenetic and population genetic results. If I understand the two data graphics, *all* of the individual fish had intermediate probability of assignment to one or the other of the k=2 groups (in each of the data graphics, every fish has a large blue region and a large orange region), and none of the fish was assigned with a high probability to either group (which would be shown in the graphic by a bar with mostly blue or mostly orange). This is a common result for STRUCTURE analyses where the model is overspecified and the number of groups in the analysis (k=2) is larger than the actual number of biological groups (k=1 in the case of Gila robusta).

I don't think that's necessarily a reason for the authors to add the STRUCTURE analysis and result to a second revised manuscript, but it seemed worthwhile to add that comment (and encourage the authors to follow up with more data if they are not going to publish these population genetic analyses now).

Reviewer 2 ·

Basic reporting

A very nicely written manuscript that I believe has improved from the original submission. That taxonomic confusion with little basis for differentiation among taxa reigns historically is clearly presented and that presentation is without bias, making the manuscript rather neutral - this is a strong point of the submission. The presentation is eminently fair and thorough. The authors' contention that they sacrificed the data in lieu management (e.g., offered the data in a symposium) is well taken and satisfies the criticism that nothing new is presented here.

Experimental design

Design is solid, no issues have arisen since the first submission. The inclusion of the coalescence analyses is a strength.

Validity of the findings

Strong and obviously supported by the morphological and molecular data presented.

Additional comments

Nice job dealing with a politically charged issue. The fairness by which the data and conclusions are presented make the manuscript rather neutral, and that is a strength.

---

## Round 0.3 · accepted · Accept

Dear Authors,

Thank you for the revision. I now deem the MS acceptable for publication. However, I would still appreciate if you could report ln likelihoods of one and two biological groups in the Structure analysis.

Congratulations.

Tomas Hrbek

#